# A Multiband Shared Aperture MIMO Antenna for Millimeter-Wave and Sub-6GHz 5G Applications

**DOI:** 10.3390/s22051808

**Published:** 2022-02-25

**Authors:** Rifaqat Hussain, Mohamed Abou-Khousa, Naveed Iqbal, Abdullah Algarni, Saad I. Alhuwaimel, Azzedine Zerguine, Mohammad S. Sharawi

**Affiliations:** 1Electrical Engineering Department, King Fahd University for Petroleum and Minerals (KFUPM), Dhahran 31261, Saudi Arabia; rifaqat@kfupm.edu.sa (R.H.); naveediqbal@kfupm.edu.sa (N.I.); algarnia@kfupm.edu.sa (A.A.); azzedine@kfupm.edu.sa (A.Z.); 2Department of Electrical Engineering and Computer Science, Khalifa University of Science and Technology, Abu Dhabi 127788, United Arab Emirates; mohammed.aboukhousa@ku.ac.ae; 3King Abdulaziz City for Science and Technology, Riyadh 11442, Saudi Arabia; 4Center for Communication Systems and Sensing, King Fahd University of Petroleum & Minerals, Dhahran 31261, Saudi Arabia; 5Electrical Engineering Department, Polytechnique Montréal, Montreal, QC H3T 1J4, Canada; mohammad.sharawi@polymtl.ca

**Keywords:** shared aperture, sub-6GHz, millimeter-wave, 5G, slot antenna, MIMO

## Abstract

A shared aperture 2-element multiple-input-multiple-output (MIMO) antenna design for 5G standards is presented in this study, one which uses the same radiating structure to cover both the sub-6GHz and millimeter-wave (millimeter-wave) bands. The proposed antenna comprises four concentric pentagonal slots that are uniformly separated from one another. For the sub-6GHz band, the antenna is excited by a single open-end microstrip transmission-line, while a 1 × 8 power divider (PD) connected via a T-junction structure excites the millimeter-wave band. Both the sub-6GHz and mm-wave antennas operate in a MIMO configuration. The proposed antenna design was fabricated on a 120 × 60 mm^2^ substrate with an edge-to-edge distance of 49 mm. The proposed sub-6GHz antenna covers the following frequency bands: 4–4.5 GHz, 3.1–3.8 GHz, 2.48–2.9 GHz, 1.82–2.14 GHz, and 1.4–1.58 GHz, while the millimeter-wave antenna operates at 28 GHz with at least 500 MHz of bandwidth. A complete antenna analysis is provided via a step-by-step design procedure, an equivalent circuit diagram showing the operation of the shared aperture antenna, and current density analysis at both millimeter-wave and sub-6GHz bands. The proposed antenna design is also characterized in terms of MIMO performance metrics with a good MIMO operation with maximum envelop correlation coefficient value of 0.113. The maximum measured gain and efficiency values obtained were 91% and 8.5 dBi over the entire band of operation. The antenna is backward compatible with 4G bands and also encompasses the sub-6GHz and 28 GHz bands for future 5G wireless communcation systems.

## 1. Introduction

5G is the new standard adopted for cellular-based wireless communications. It is a major leap beyond the 4G network, as it offers many features which were not available before. In particular, it offers massive connectivity, low latency, high reliability, and much higher data rates than its predecessors. The new architectures of 5G will require major changes in hardware implementation when it is deployed. Unlike the previous standards, 5G systems, will also use millimeter-wave bands for the first time [1]. This is in addition to the classical bands used in the previous generation of wireless communication, which mainly lie between 700~MHz to 6~GHz. 5G systems will rely on the sub-6GHz bands for wider coverage, while it is envisioned that millimeter-wave bands will be used in small cells where communication will take place in close proximity with extremely high data rates. Slot-based antenna designs are potentially good candidates because of their planar structure, easy integration with other circuit components, and suitability for multi-band operation, not to mention the concept of the shared aperture being relatively easy to implement when compared to other types of antenna designs.

The adoption of frequency bands which are spaced widely apart has posed specific challenges in the design of the radio frequency (RF) front-end, especially in relation to the antenna. Most of the devices connected with the cellular system have a small form factor. This is particularly true for internet-of-things (IoT)- based devices, which most of the time have very a small size. An antenna design which can work for all these bands, or having multiple antennas covering individual bands, will require due consideration to meet the required radiation characteristics while easily fitting inside a given device.

Multiple-input-multiple-output (MIMO) systems are an important technological breakthrough which can offer better channel capacity and reliability of communication within a limited bandwidth and transmission power. They have been adopted in several communication standards, including the 4G standard and WiFi, etc. MIMO systems will remain important in the context of the 5G standard as well [1]. In the 5G sub-6GHz bands, the adoption of the MIMO system will provide the best combination of coverage and channel capacity. In the millimeter-wave band, when the exact location of the communication device will not be known, MIMO will prove helpful by providing better channel capacity in a rich multi-path environment, where the transmitter and receiver will be in close proximity to each other.

The antenna is a key component of a MIMO system, and its design has a significant impact on the overall system’s performance. Ideally, such an antenna is required to possess very high port isolation between its antenna elements while the radiation patterns of its elements are such that the MIMO channels are kept independent. The envelop correlation coefficient (ECC) is the performance metric of a MIMO antenna which determines how much correlation between channels is added as a result of the antenna. Practically, such a MIMO antenna is designed to have high port isolation between its antenna elements and low ECC values [2].

Lately, several MIMO antenna designs have appeared in the literature for 5G systems. They can be broadly categorized into two categories based on their frequency of operation. Many designs are only for the sub-6GHz band of 5G while other designs focus on the millimeter-wave bands. In the sub-6GHz bands, the design of MIMO antennas of up to ten elements can be found in the literature for handheld devices. The focus of many of the designs is to achieve multi-band or wideband characteristics with improved isolation in closely placed antenna elements. In [3], an 8-element MIMO antenna for 5G application was designed which operates in the 2.6~GHz band. The elements are comprised of two different types of radiating structures. The first set consists of four L-shaped monopole antennas, while the second set has four C-shaped coupled-fed antennas. The antenna was designed on an 68 × 136 mm^2^ FR4 substrate with a thickness of 1~mm. With a dual polarized operation, the antenna achieved a reasonable isolation of 12.5 dB between its antenna elements. In [4], a multi-band MIMO antenna was designed which operates in the 4G bands and covers the 3.4–3.6 GHz 5G band. The 5G band is covered by four slot antenna elements. In the 5G bands, the isolation between antenna elements is significantly larger than 13 dB. The MIMO antenna presented in [5] operates in the 3.5 GHz 5G band and has eight antenna elements. Designed on an FR4 substrate, it occupies an 150 × 80 mm^2^ area. The metal rim of a mobile phone was utilized to create an 8-element wideband MIMO antenna in [6]. The 8-elements are a combination of four dual-antenna pairs, providing dual polarized operation. The antenna operates from 3.3~−5~GHz and provides a minimum isolation of 12 dB between the antenna elements. A wideband MIMO antenna comprised of 4-element open-slot antennas was proposed for 5G applications in [7]. The antenna works from 3.3–−4.2 GHz. Designed on an FR4 substrate, the antenna occupies an area of 42 × 42 mm^2^. Similarly, a MIMO antenna design for 5G smartphone applications which covers the 3.5~GHz 5G band can also be found in [8]. Broadband MIMO antennas are presented in [9,10] where a fork-shaped isolation structure and amplifier integration are utilized for isolation improvement which covers both 4G and sub-6GHz 5G bands.

The antennas designed for 5G applications covering the millimeter-wave band are mainly focused on having an array with high gain and highly directive beams. Many of these designs are antenna arrays in which multiple radiating elements are used together to achieve the desired radiation characteristics. In [11], an array antenna made up of 8-element patch antennas was excited by a power divider (PD) designed using substrate integrated waveguide (SIW). The antenna operates at 28 GHz and has a gain of 13.97 dBi. A millimeter-wave antenna for 5G communication is presented in [12]. The design consists of a dual-feed square loop antenna which is made at two millimeter-wave frequencies of 38 GHz and 75 GHz. The antenna has a gain of 3~dBi and a fractional bandwidth of 14%. In [13], a patch antenna array is presented consisting of 16 antenna elements. The antenna operates from 26 GHz to 31.4 GHz and provides a dual polarized operation. The antenna beam can be switched based on the feeding method and provides the beam in two directions with a gain of 12 dBi. A modified patch antenna with E-shape is presented for 5G millimeter-wave communication in [14] and operates in the 38 GHz band of 5G. Similarly, Ref. [15] describes a high gain antenna for 5G. The antenna operates in the 28 GHz band with a gain of 23 dBi. Mutual coupling is an important MIMO antenna design parameter. For the proposed antenna design with sufficient spacing, no mutual coupling reduction structure was incorporated. However, various techniques, such as a complementary split ring resonator, defected ground structure, electronic bandgap structure, parasitic or slot element, and decoupling networks, have been reported in literature [16,17,18].

In addition to designs which incorporate arrays and single elements for arrays to achieve high gain in the millimeter-wave band of 5G, some MIMO antennas for 5G in the millimeter-wave band have also appeared in literature [19,20]. In [19], a quasi Yagi-Uda like array of a 4-element antenna was designed which operates at 36 GHz. A decoupling method for isolation improvement in a DRA-based MIMO antenna operating in the millimeter-wave band is presented in [21]. Some recent MIMO antenna designs have also appeared which cover both the sub-6GHz band and the millimeter-wave band simultaneously [22,23]. All such designs use separate radiating elements for the sub-6GHz band and the millimeter-wave band. In this work, a new MIMO antenna is proposed for 5G systems which covers the sub-6GHz/millimeter-wave bands. The same radiating structure is used to excite widely space bands. Consisting of two antenna elements, each element is a modified slot antenna which is excited by microstrip transmission-lines.

This paper proposes a novel 2-element MIMO antenna design to be used for 5G-enabled access point devices, in which the same elements operate in the sub-6GHz and millimeter-wave bands, although with different feeding structures. The contributions of the proposed work are:The distinctive characteristic of the proposed antenna design is the integration of sub-6GHz with a millimeter-wave antenna that uses the same radiating aperture/structure.This study provides a unique concept for 5G access point applications based on a shared-aperture sub-6GHz and millimeter-wave concentric pentagonal slots-based antenna design. It eliminates the need for multiple antenna structures to be used for different operating bands, namely the sub-6GHz and millimeter-wave bands.The multi-band antenna design proposed in this work operates on several fundamental modes corresponding to each concentric slot. As a result, the antenna appealing due to the higher gains and efficiency values attained for every dominant mode.The MIMO performance of the proposed antenna design is well-suited to be operated in both the sub-6GHz and 28 GHz millimeter-wave bands.Despite their distinct feeding structures, the proposed design employs a co-shared radiating structure covering the desired sub-6GHz and millimeter-wave bands. This is one of the distinguishing traits of the proposed antenna design.The covered bands for sub-6GHz are 4–4.5~GHz, 3.1–3.8~GHz, 2.48–2.9~GHz, 1.82–2.14~GHz, and 1.4–1.58~GHz, whereas the millimeter-wave ranges from 27.8 GHz to 28.3 GHz with a minimum bandwidth (BW) of 500 MHz. The operation of such a hexa-band MIMO antenna for both sub-6GHz and millimeter-wave bands has not been reported for any antenna previously.

The paper is organized as follows. In Section 2, details of the antenna design are provided along with the design procedure and the equivalent circuit model. In Section 3, the measured and simulated results are comprehensively provided along with the radiation characteristics of the proposed antenna design. Future directions and the conclusions of the proposed work are provided in Section 4 and Section 5, respectively.

## 2. Antenna System Design

In this section, details about the geometry of the antenna, a circuit diagram, the theory behind the operation of the antenna, and the design procedure are discussed in detail.

### 2.1. Antenna Geometry

Figure 1 depicts the geometry of the designed shared aperture slot-based MIMO antenna. The antenna proposed in this work was designed on a Rogers RO-4350 substrate with a thickness of 0.508~mm and a dielectric constant (*ε_r_*) of 3.48. The total size of the board used was 60 × 120 mm^2^. Each antenna element is comprised of four concentric pentagonal slots etched out from the ground (GND) plan. The gap between the slots and the width of each slot are 2 mm and 0.8 mm, respectively. The optimal spacing between the antenna elements was found to be 70 mm. The proposed slot-based shared aperture design was excited with an open-ended transmission-lines as well as with 1 × 8 PD. The transmission-lines (Feed-1 and Feed-2) and PD (Feed-3 and Feed-4) were used to excite the same slot structure for sub-6GHz bands and millimeter-wave band at 28 GHz, respectively. Feed-1 and Feed-2 were regular SMA connectors which were utilized for sub-6GHz operation, while Feed-3 and Feed-4 were PCB receptacle 2.92 mm F type connectors that cover up to 40 GHz. The fabricated model of the proposed antenna is shown in Figure 2, while the various antenna dimensions are given in Table 1.

### 2.2. Antenna Design Procedure

The shared aperture slot antenna proposed in this work was designed to cover as many sub-6GHz 5G bands as possible as well as the millimeter-wave band at 28 GHz. For the proposed work, a pentagonal shaped slot was selected due to the fact that it can achieve multi-band frequency characteristics and better input impedance matching compared to annular slot antennas [24]. Additionally, the open-ended microstrip line feed was chosen for sub-6GHz and millimeter-wave frequency bands because of its excellent reactance compensation effect, which offers an improved flatness of the frequency response and hence an improved operational bandwidth [25,26].

To achieve the lowest possible frequency bands, a single pentagonal slot was initially chosen. A single slot antenna with resonance curves, depicted in Figure 3a, was simulated. It was apparent that the slot on the far side was running at 1.5 GHz. Inner slot-line structures were used to obtain upper frequency bands in the same way. The *S*_11_ plots for the two and three concentric pentagonal slots are shown in Figure 3b,c, respectively.

In order to determine the antenna’s dimensions, its resonating bands, together with adequate input impedance matching and proper placement on the board, a number of parametric sweeps were performed. To achieve the desired performance, the width of each pentagonal slot and the spacing between them was optimized. The *S*_11_ curves for optimized design are depicted in Figure 3d.

A 50 Ω microstrip line was used to feed the pentagonal slot-line structure. The standard microstrip line equation was used to calculate the feed line width. Each pentagonal slot resonated at its fundamental and higher order modes. The four concentric slots were tweaked to cater for as many sub-6GHz 5G bands as possible. This was accomplished by carefully analyzing the proposed antenna’s *Z_in_*. The length and width of the feed line were optimized in order to match the input impedance *Z_in_*. For sub-6GHz operation, parametric sweeps were performed on the length of the feed line (see Figure 4). It was discovered that at the resonating bands, the real part of *Z_in_* is around 50 Ω, whereas the imaginary part of *Z_in_* is near to zero for the optimized design.

A 1 × 8 PD was designed at 28 GHz, as shown in Figure 5a. To feed the shared aperture slot structure, a standard power dividing mechanism was adopted. The proposed PD arrangement was used to feed the slot antenna structure periodically. At the required millimeter-wave frequency band, the spacing between consecutive feed branches was fixed at λ_g_/2. As a result, the slot planar connected antenna array configuration was fed by the PD. The PD current density plot at 28 GHz is shown in Figure 5b. It is evident that the input port transmitted an equal amount of power for the desired millimeter-wave band. The millimeter-wave antenna is a part of the sub-6GHz structure. The pentagonal slots are excited using microstrip-transmission lines. The effective portion of the slots radiating in the millimeter-wave band is shown using the current density figures. Figure 5c shows the current density when the millimeter-wave port is excited while the other ports are terminated with 50 Ω. It can be seen that energy is coupled to the outermost slot while both the millimeter-wave ports are well isolated. Figure 6a,b depict the reflection curves *S*_11_ at the input port (*P_in_*) and the transmission curves *S*_1*n*_, respectively, where *n* = 2, 3,…, 9, denotes the number of output ports (*P_o_*_1_, *P_o_*_2_…*P_o_*_8_) of the PD. It can be observed that between the frequencies of 27 GHz and 29 GHz, the input port transmitted an equal amount of power. This is the millimeter-wave antenna’s desired operating band.

To obtain the desired PD characteristic, numerous parametric sweeps were carried out. Figure 7a shows the variation in the length of PD branches coupled with the slot elements, while the corresponding *S*_11_ are shown in Figure 7b. The PD lengths were varied to obtain the optimal performance of the millimeter-wave band antenna. It was observed that the outermost pentagonal slot was acting as a connected array for the millimeter-wave band while the inner slot was not part of the radiating structure. This was verified from the current distribution at 28 GHz, as shown in Figure 5c. The optimized sub-6GHz antenna was also simulated with and without the millimeter-wave PD, as shown in Figure 8a. Similarly, the millimeter-wave antenna was also simulated to determine the effect of the microstrip feed line on the *S*_33_. A slight variation in scattering parameters was observed for both cases, as shown in Figure 8.

To elaborate on the antenna’s operation, although the two antennas at the sub-6GHz and millimeter-wave bands share the same aperture, they are excited by a different set of transmission lines. Thus, the effective aperture of the two antennas is different and the millimeter-wave antenna is not of a higher order in terms of mode compared to the sub-6GHz antenna. We investigated further by simulating the proposed design while Feed-1 and Feed-2 were active with a frequency sweep from 27–29 GHz. The resulting *S*_11_ curves are shown in Figure 8c. The antenna did not resonate at 28 GHz with the micro-strip feed line. Hence, it can be concluded that the millimeter-wave PCA operates at its fundamental mode of operation thanks to the proper design of the PD and slot-connected array.

### 2.3. Antenna Operation

The planar slot-based antenna design is an attractive choice for researchers because of its numerous advantages. Both ends of an open-ended and short-circuited slot-antenna, respectively, can be modeled as λ/4 and λ/2 transmission lines, corresponding to the resonating band fundamental mode [27], as given by:(1)$$f_{r}=\frac{c}{\pi (l_{2}+l_{1})}\times \sqrt{\frac{\varepsilon_{r}+1}{2\varepsilon_{r}}}$$
where $\varepsilon_{r}$ is the substrate board’s relative permitivity, c is the speed of light, $f_{r}$ is the fundamental resonance frequency, and $l_{1}\;\&\;l_{2}$ are the inner and outer average slots lengths, as shown in Figure 1 [28].

The circuit model of the square slot-line sub-6GHz radiator is shown in Figure 9. The equivalent circuit was modeled as a parallel combination of an *RLC* circuit to aid in the understanding of the shared aperture slot structure for the sub-6GHz and millimeter-wave bands. A microstrip slot-line transition represents an interesting option for the slot antenna feeding mechanism because of its simple planar structure, efficient coupling with the slot, and well-known behavior [26]. The circuit model of such a feeding transition was reported in [29]. In the sub-6GHz equivalent circuit model, $C_{f}$ and $L_{f}$ are the capacitance and inductance of input microstrip feed line. It also included transformer coupling with a series capacitance $C_{om}$, representing an open-ended circuit capacitance of the microstrip feed. The secondary circuit included a shunt reactance represented by $L_{i}(i=1,2,3,4)$, which is at the end inductance of the slot-line [25,29]. The proposed four concentric slot-based antenna design has five sub-6GHz operation bands, with each slot resonating at the fundamental mode of operation, while the last band is linked to the first higher order modes of the largest slot-line structure.

From the design evolution process, it can be seen that each slot-line did not affect the previous resonating band. Each slot in the proposed sub-6GHz antenna structure operates at its fundamental mode of operation (see Figure 3a–c), resulting in three distinct resonating bands (Band-1, Band-2, and Band-5). However, the second inner most slot resonates at its fundamental mode (Band-3), and the superposition of two inner slots results in a fourth resonating band (Band-4). In the equivalent circuit model, each slot is acts as a separate parallel *RLC* circuit, as shown in the equivalent circuit model, while the inner two slots are also inductively coupled with each other. Similarly, for the millimeter-wave band antenna, the same antenna aperture was shared and acts as a connected slot array fed with Port-3. This can be modeled as a parallel resonating *RLC* circuit, as shown in Figure 9. To comprehend the antenna multi-band, equivalent circuit parameters, and shared aperture operation, an equivalent circuit model, together with analysis, has been provided.

## 3. Results and Discussions

The proposed shared aperture MIMO slot antenna was designed and modeled using HFSS^™^. Both the sub-6GHz and millimeter-wave bands were tested with the optimized antenna design. In this section, the measured and simulated S-parameters, gain patterns, and MIMO performance parameters are discussed in detail. The port parameters were obtained using an Agilent PNA, while a Satimo Starlab chambers were used for measurement of patterns.

### 3.1. Scattering Parameters for MIMO Antenna

The measured and simulated S-parameters of the proposed antenna design are presented in Figure 10. Once Feeds 1 and 2 are excited, the proposed antenna design acts like a multi-bands antenna resonating at: 4–4.5~GHz, 3.1–3.8~GHz, 2.48–2.9~GHz, 1.82–2.14~GHz, and 1.4–1.58~GHz. The given sub-6GHz bands covered most of the 5G standard, in addition to the 4G and GPS bands. Figure 10a presents the measured and simulated reflection coefficient curves of the sub-6GHz bands. The minimum isolation obtained between the two antenna elements was around 15~dB, as shown in Figure 10b. The two antenna elements were well isolated over all the resonating bands.

Similarly, the activation of Feeds 3 and 4 causes the antenna to function in the millimeter-wave band. The reflection coefficient when these feeds are excited is shown in Figure 10c. As a result, the antenna operates between 27.8 and 28.3 GHz, covering the 28 GHz 5G band. As shown in Figure 10d, the isolation between the antenna elements in this band is greater than 25 dB. To analyze the mutual coupling between antenna elements for sub-6GHz operation, the current density distribution was analyzed at 4.25, 3.6, 2.69, and 1.98 GHz, as shown in Figure 11a–d, respectively. Low mutual coupling values were observed for the given frequency bands. Such analyses are important when it comes to designing a defected ground structure to enhance the isolation between closely spaced antenna elements. The electrical distance between antenna elements is smaller at sub-6GHz when compared to the millimeter-wave band. Hence, low isolation levels were obtained at sub-6GHz and high values were observed for the millimeter-wave band.

### 3.2. Radiation Characteristics

The shared aperture MIMO antenna proposed in this work was characterized in terms of its far-field radiation patterns. The efficiency (%η) and peak gain values were investigated for the sub-6GHz/millimeter-wave bands. For each measurement, all the ports were terminated with a 50 Ω load except the port being measured. Both measured and simulated peak gain (PG) and efficiency (%η) values are listed in Table 2.

Figure 12 depicts the 2D gain patterns as well as the anechoic chamber measurement setup, presented in Figure 12e. The normalized gain patterns at 1.9 GHz for antenna elements Ant-1 and Ant-2 are presented in Figure 12a,b, respectively. The curves are θ-cut at $\phi =0^{{}^{\circ}}$ and $\phi =90^{{}^{\circ}}$. Similarly, the 2-D curves for Ant-1 and Ant-2 at 4.2 GHz are shown in Figure 12c,d, respectively. For the millimeter-wave band operating at 28 GHz, the normalized gain patterns for antenna elements Ant-1 and Ant-2 are shown in Figure 13a,b, respectively. The curves are θ-cut at $\phi =0^{{}^{\circ}}$ and $\phi =90^{{}^{\circ}}$. Both measured and simulated peak gain (PG) and efficiency (%η) are shown in Figure 14a–d.

### 3.3. Radiation Characteristics with Reflector

A slot antenna’s radiation pattern is not unidirectional, radiating in both forward and backward directions. For antennas that communicate via line-of-sight (LOS) and where the precise location of the receiver and transmitter are known, directional patterns are highly desirable. However, in a close proximity communication scenario, where the location of the receiver and transmitter are not known, MIMO antennas are more suitable, especially in a scattering rich environment [28,29].

The radiation pattern at 28 GHz is not directional, which was expected because of the slot-based antenna design. A simple modification was made for this design to reduce the ripples and make the patterns more directional by using a backing reflector, which is usually present because of the electronics behind the antenna in access point applications. A simulation was performed to show the patterns at 28 GHz. Figure 15 shows the more directional patterns with minimum ripples in the presence of such a reflector with a height h=3 mm. The reflector also helped to minimize the backward radiation at sub-6GHz bands, and therefore more directional patterns were obtained.

### 3.4. MIMO Performance Parameters

The correlation coefficient (ρ) is a critical MIMO antennas parameter to find the correlation between various channels. It is the square root of envelop correlation coefficient (ECC) ($\rho_{e}$). ECC can be computed using the far field radiation characteristics of antennas, as given below [2]:(2)$$\rho_{e}=\frac{{\left|\iint G_{1}(\theta ,\phi )*G_{2}(\theta ,\phi )d\Omega \right|}^{2}}{\iint {\left|G_{1}(\theta ,\phi )\right|}^{2}d\Omega \iint {\left|G_{2}(\theta ,\phi )\right|}^{2}d\Omega }$$
where $G_{i}$ represents the radiation pattern when element i is active. The maximum acceptable $\rho_{e}$ value for proper MIMO operation is 0.5, which corresponds with correlated channels, while the minimum value corresponds with the highly uncorrelated channel.

ECC values were computed for the sub-6GHz/millimeter-wave bands and are shown in Table 2. It is critical to evaluate the MIMO performance using ECC ($\rho_{e}$), which uses radiation patterns to determine the field coupling between various correlated channels. For improved MIMO operation, the $\rho_{e}$ value must be less than 0.5. ECC values of less than 0.15 were observed across the entire operating bands (sub-6GHz and millimeter-waves), implying good MIMO performance.

To assess the proposed shared aperture MIMO antenna system’s diversity performance, diversity gain (*DG*) values were computed. In a MIMO system, higher values of signal to noise ratio (*SNR*) are obtained for uncorrelated signals and hence better signal reception is provided. Diversity gain is the difference between the time-averaged *SNR* of the given MIMO diversity antenna system and that of a diversity channel of a single antenna system with the same *SNR* reference level. Mathematically,
(3)$$DG={\left[\frac{\gamma_{c}}{SNR_{c}}-\frac{\gamma_{1}}{SNR_{1}}\right]}_{P(\gamma_{c}<\gamma_{s}/SNR)}$$
where $SNR_{c}$ and $\gamma_{c}$ are the mean and instantaneous SNRs for a MIMO diversity system, while $SNR_{1}$ and $\gamma_{1}$ are the mean and instantaneous SNRs for a single channel diversity system, respectively. Furthermore, $\gamma_{c}/SNR_{c}$ are the reference level [2]. The obtained *DG* values for sub-6GHz and millimeter-wave bands are given in Table 2.

The S-parameter can be used to obtain resonance and bandwidth information for MIMO antenna systems, but it does not provide a frequency response, especially in relation to effective bandwidth. The total active reflection coefficient (TARC) can be used to characterize the frequency response of a MIMO antenna effectively. This can be defined as the ratio of the square root of the total reflected power to the square root of the total incident power and is given by expression blow:(4)$$\Gamma_{a}^{t}=\frac{\sqrt{\sum_{i=1}^{N}{\left|b_{i}\right|}^{2}}}{\sqrt{\sum_{i=1}^{N}{\left|a_{i}\right|}^{2}}}$$
where $a_{i}$ and $b_{i}$ are the incident and reflected signals, respectively.

For the proposed shared aperture MIMO antenna design, TARC curves for sub-6GHz/millimeter-wave bands were obtained. The phase of the first exciting port was kept at 1e^j0^, while this varied from $0^{{}^{\circ}}$ to $210^{{}^{\circ}}$ for the second port. The TARC curves for sub-6GHz and millimeter-wave bands are shown in Figure 16a,b, respectively. From the given curves, it is clear that the effective bandwidth was quite robust for large deviation in the excitation phases of the input ports. Details of five different combinations of phases for both bands are given below:**Case-1:**$P-1=P-3=0^{{}^{\circ}};\;P-2=P-4=30^{{}^{\circ}};$**Case-2:**$P-1=P-3=0^{{}^{\circ}};\;P-2=P-4=60^{{}^{\circ}};$**Case-3:**$P-1=P-3=0^{{}^{\circ}};\;P-2=P-4=70^{{}^{\circ}};$**Case-4:**$P-1=P-3=0^{{}^{\circ}};\;P-2=P-4=140^{{}^{\circ}};$**Case-5:**$P-1=P-3=0^{{}^{\circ}};\;P-2=P-4=210^{{}^{\circ}};$

The mean effective gain (*MEG*) accounts for the antenna coverage area and the achievable data rates of the MIMO antenna system. *MEG* can be calculated via the mutual relation between the field patterns of the antenna and the statistical distribution of signals in the urban environment [2]. The *MEG* of an antenna system is determined from its two-dimensional field patterns in a mobile wireless environment.
(5)$$MEG=\int_{0}^{2\pi }\left[\frac{XPD}{1+XPD}E_{\theta }+\frac{XPD}{1+XPD}E_{\phi }\right]d\phi$$
where XPD is the cross polarization discriminator. The value of *MEG* is −3 dB for urban environments with a 100% efficient antenna, while for practical designs, the value can be as low as −12 dB. The *MEG* of the proposed MIMO antenna was computed for sub-6GHz/millimeter-wave bands using Equation (5) and the values are provided in Table 3 for different *XPD*.

Table 4 presents a thorough comparison between the proposed MIMO antenna and existing state-of-the-art works. Antenna arrays for sub-6GHz and millimeter-wave are integrated in the given antenna design. In comparison to other related works, it is evident that the designed antenna excels in numerous ways.

## 4. Future Directions

This work presents a 2-element shared aperture MIMO antenna design for 5G-enabled devices and access point applications. This work can be extended by incorporating 4-elements within the same aperture with polarization diversity and enhanced gain. The shared aperture concept can also be implemented in mobile terminals and other wireless handheld devices.

## 5. Conclusions

This work has presented a 2-element shared aperture MIMO antenna design for 5G-enabled devices and access points. The design covered the sub-6GHz and millimeter-wave bands with the same radiating elements, and four concentric pentagonal slots comprised the single element. To excite the antenna for the sub-6GHz and millimeter-wave bands, a single open-end microstrip transmission line and a 1 × 8 PD, respectively, were used. The proposed antenna design was fabricated on a 120 × 60 mm^2^ substrate board. The proposed sub-6GHz antenna covers the following frequency bands: 4–4.5~GHz, 3.1–3.8~GHz, 2.48–2.9~GHz, 1.82–2.14~GHz, 1.4–1.58~GHz, and 27.8–28.3~GHz, while the millimeter-wave antenna operates at 28 GHz with a minimum bandwidth of 1 GHz. The simulation results were in close agreement with the measured results. The antenna is not only backward compatible with 4G bands, but it also has coverage for 5G sub-6GHz and 28 GHz bands.

## Figures and Tables

**Figure 1 sensors-22-01808-f001:**
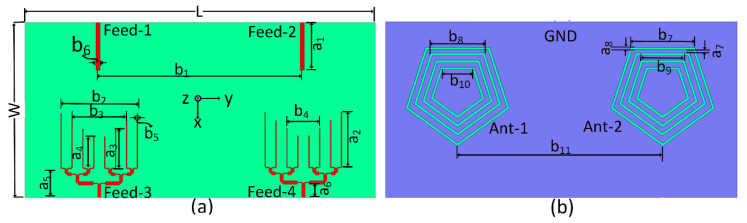
Proposed MIMO antenna: (**a**) Top layer, (**b**) GND plane.

**Figure 2 sensors-22-01808-f002:**
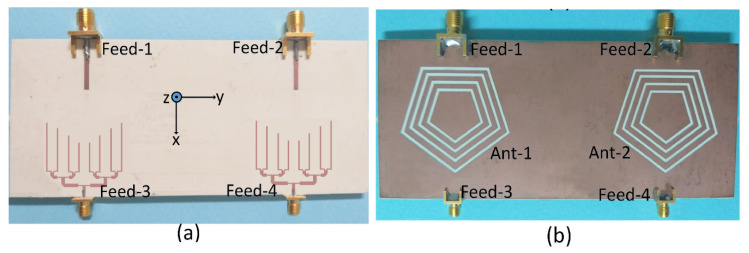
Fabricated antenna: (**a**) Top view, (**b**) bottom view.

**Figure 3 sensors-22-01808-f003:**
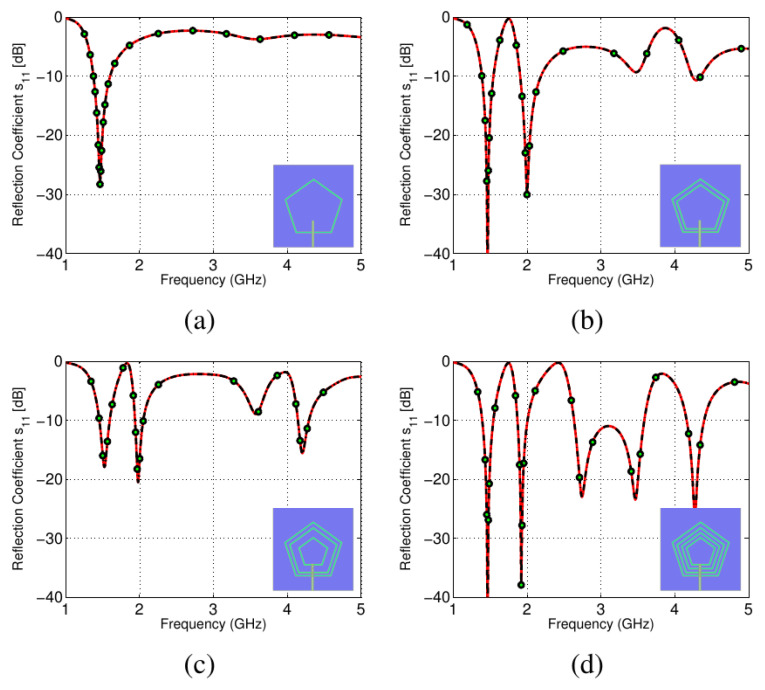
Steps in the design process: (**a**) *S*_11_ with one slot, (**b**) *S*_11_ with two slots, (**c**) *S*_11_ with four slots, (**d**) *S*_11_ with an inner pentagonal plate slot.

**Figure 4 sensors-22-01808-f004:**
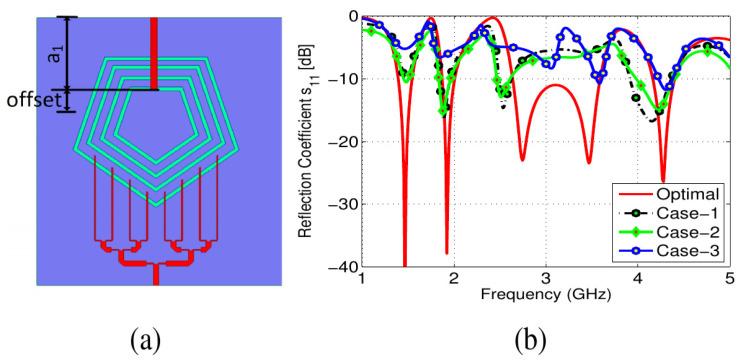
Effects of feed length: (**a**) Antenna with feeds (**b**) Optimal offset = 0 mm, Case-1: offset = 1 mm, Case-2: offset = 1 mm, Case-3: offset = 3 mm.

**Figure 5 sensors-22-01808-f005:**
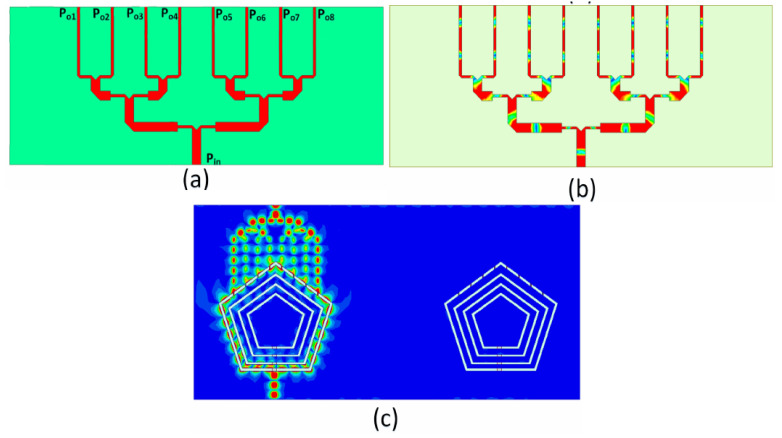
Current density distribution at 28 GHz: (**a**) PD, (**b**) CD of PD, (**c**) Ant-3 at 28 GHz.

**Figure 6 sensors-22-01808-f006:**
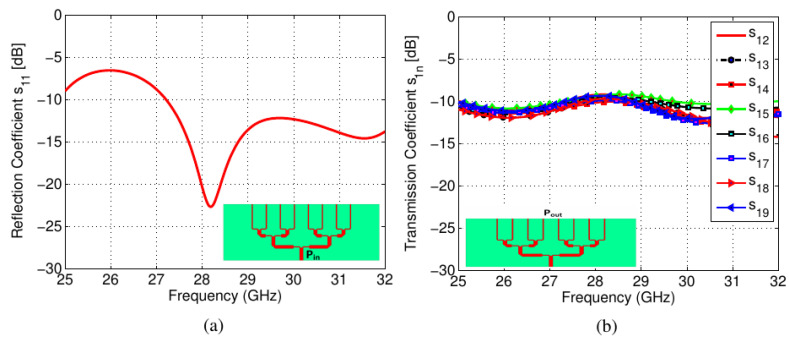
Scattering parameters of PD: (**a**) *S*_11_, (**b**) *S*_1*n*_.

**Figure 7 sensors-22-01808-f007:**
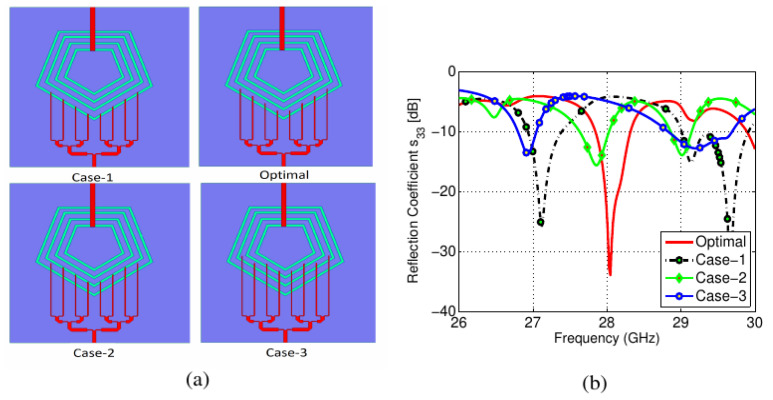
Effects of PD branch lengths: (**a**) Case-1: offset = −1 mm, Optimal case: optimal offset = 0, Case-2: offset = 2 mm, Case-3: offset = 4 mm; (**b**) *S*_11_ for all given cases.

**Figure 8 sensors-22-01808-f008:**
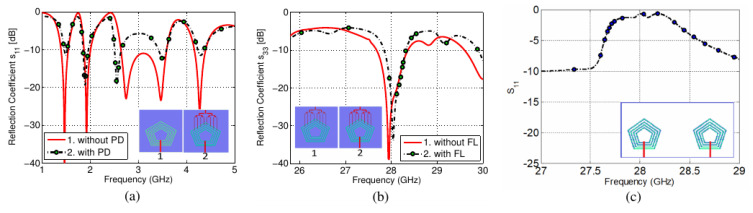
Input feed analysis: (**a**) *S*_11_ with and without PD, (**b**) *S*_33_ with and without the microstrip feed line, (**c**) Port−1 excited at 28 GHz.

**Figure 9 sensors-22-01808-f009:**
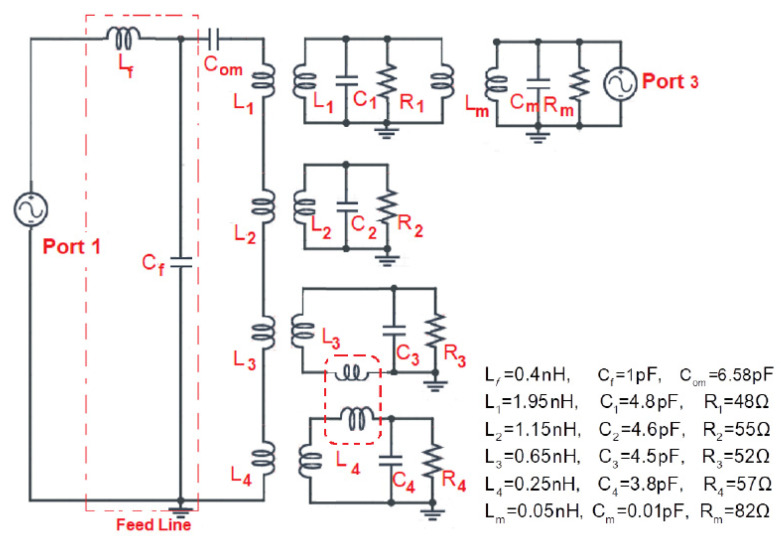
Equivalent circuit model of sub-6GHz and millimeter-wave radiator.

**Figure 10 sensors-22-01808-f010:**
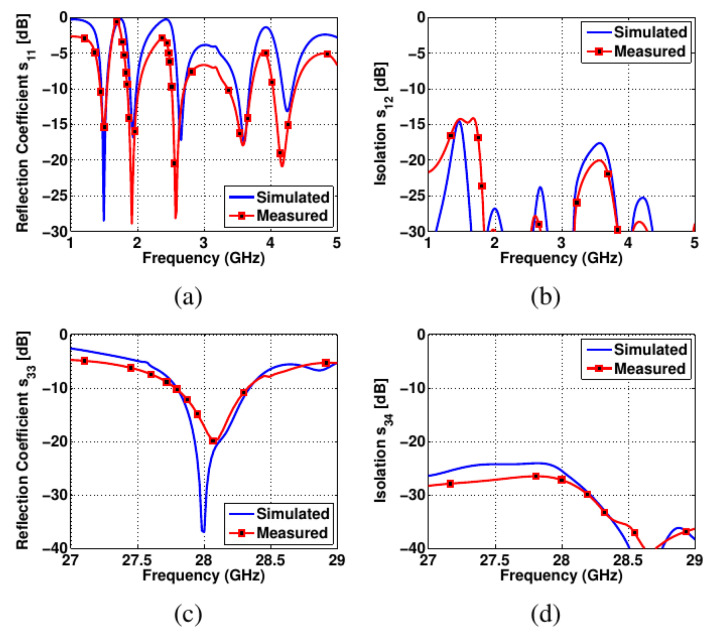
Curves showing measured and simulated values: (**a**) *S*_11_, (**b**) *S*_12_, (**c**) *S*_33_, (**d**) *S*_34_.

**Figure 11 sensors-22-01808-f011:**
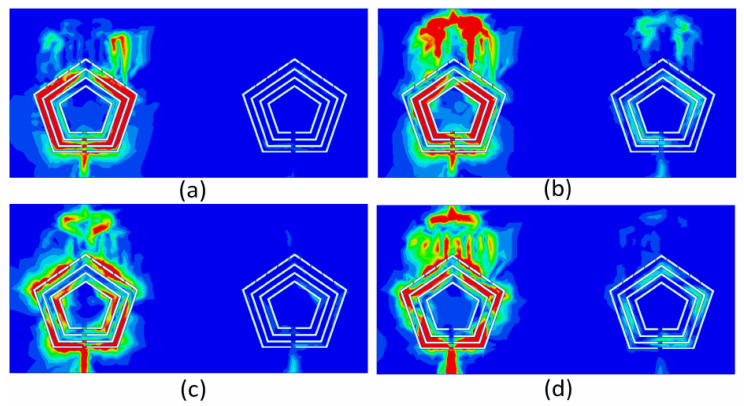
Current density distribution: (**a**) Ant-1 at 1.98 GHz, (**b**) Ant-1 at 2.69 GHz, (**c**) Ant-1 at 3.6 GHz, (**d**) Ant-1 at 4.25 GHz.

**Figure 12 sensors-22-01808-f012:**
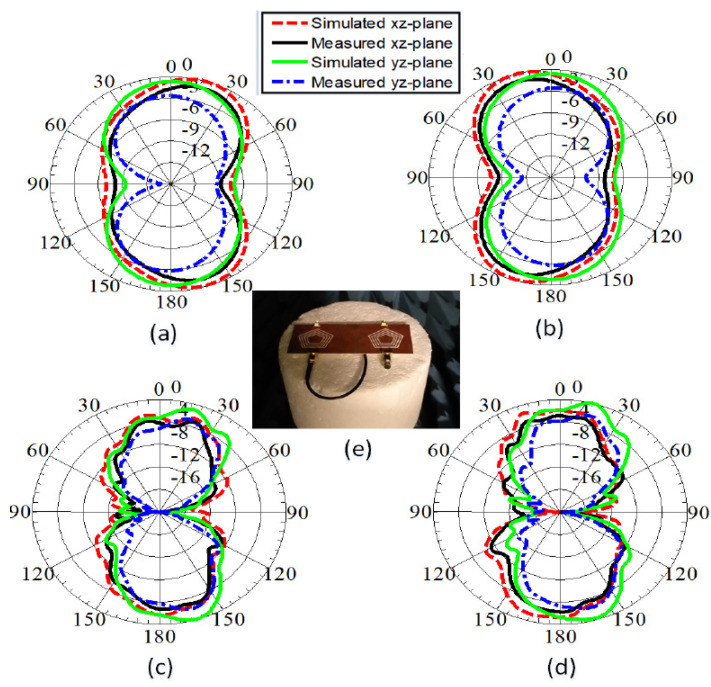
Measured and simulated gain patterns: (**a**) Ant−1 at 1.9 GHz, (**b**) Ant−2 at 1.9 GHz, (**c**) Ant−1 at 4.2 GHz, (**d**) Ant−2 at 4.2 GHz, (**e**) Measurement setup.

**Figure 13 sensors-22-01808-f013:**
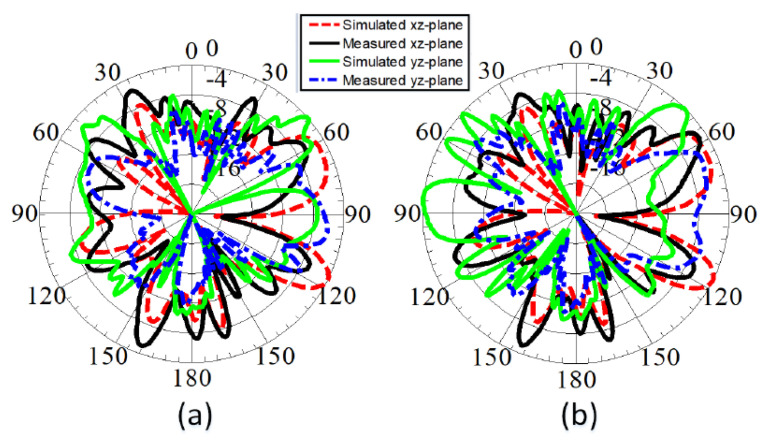
Measured and simulated gain patterns: (**a**) Ant−1 at 28 GHz, (**b**) Ant−2 at 28 GHz.

**Figure 14 sensors-22-01808-f014:**
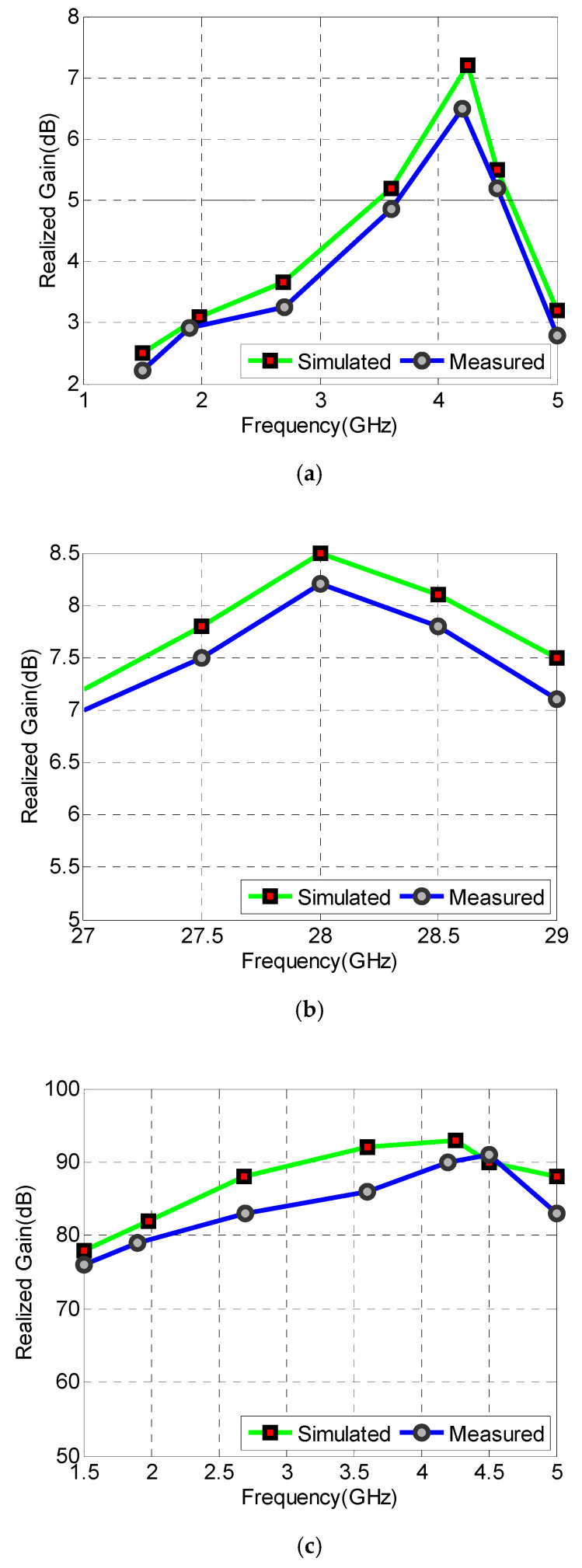
Simulated and measured curves: (**a**) Peak Gains at sub-6GHz bands, (**b**) peak Gains at mm-wave band, (**c**) %*η* at sub-6GHz bands, (**d**) %*η* at mm-wave band.

**Figure 15 sensors-22-01808-f015:**
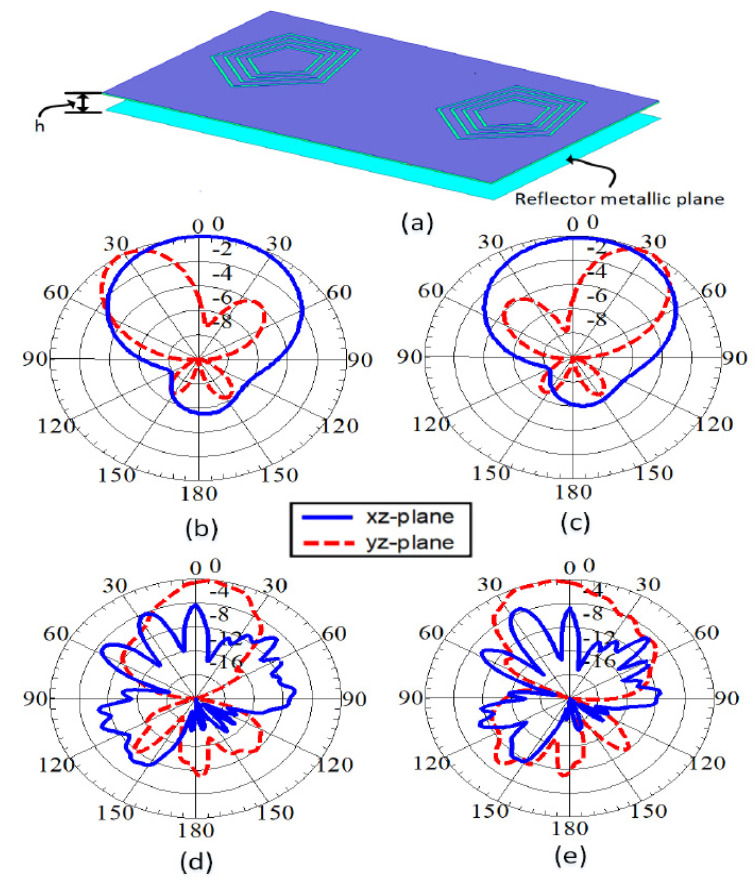
Simulated gain patterns with back-plane reflector: (**a**) Antenna with reflector, (**b**) Ant−1 at 1.9 GHz, (**c**) Ant−2 at 1.9 GHz, (**d**) Ant−1 at 28 GHz, (**e**) Ant−2 at 28 GHz.

**Figure 16 sensors-22-01808-f016:**
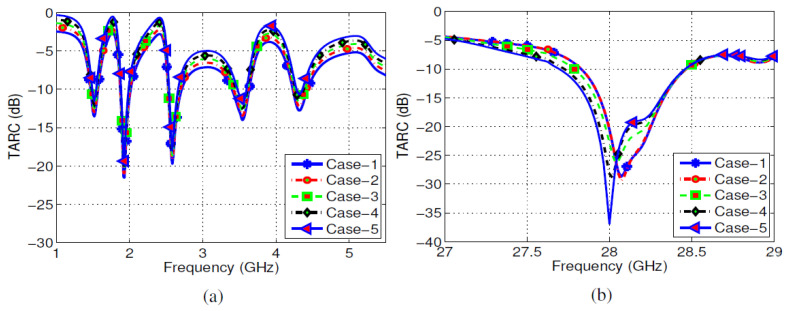
TARC curves: (**a**) sub-6GHz, (**b**) millimeter-wave band.

**Table 1 sensors-22-01808-t001:** Various antenna dimensions.

Variable	Value (mm)	Variable	Value (mm)	Variable	Value (mm)
L	120	a_6_	4.3	b_5_	0.27
W	60	a_7_	2	b_6_	1.1
a_1_	16.3	a_8_	0.5	b_7_	21.94
a_2_	19.36	b_1_	70	b_8_	18.41
a_3_	13.66	b_2_	26.42	b_9_	13.71
a_4_	10.95	b_3_	18.95	b_10_	10.77
a_5_	12.38	b_4_	11.48	b_11_	70

**Table 2 sensors-22-01808-t002:** ECC, (%η), peak gain (PG), and diversity gain (DG).

Simulated Results					Measured Results				
*f_s_* (GHz)	ρ_12_	PG (dBi)	%*η*	DG	*f_m_* (GHz)	ρ_12_	PG (dBi)	%*η*	DG
*f_s_* _1_	0.105	2.5	78	9.945	*f_m_* _1_	0.113	2.23	76	9.936
*f_s_* _2_	0.121	3.1	82	9.927	*f_m_* _2_	0.105	2.92	79	9.945
*f_s_* _3_	0.096	3.66	88	9.954	*f_m_* _3_	0.099	3.25	83	9.950
*f_s_* _4_	0.089	5.2	92	9.960	*f_m_* _4_	0.091	4.85	86	9.958
*f_s_* _5_	0.086	7.2	93	9.963	*f_m_* _5_	0.095	6.5	90	9.959
*f_s_* _6_	0.065	8.5	95	9.979	*f_m_* _6_	0.014	8.2	91	9.999

**Table 3 sensors-22-01808-t003:** MEG of proposed MIMO antenna.

Freq (GHz)	MEG_1_ (*XPD* = 0 dB)	MEG_1_ *(XPD* = 3 dB)	MEG_1_ (*XPD* = 6 dB)
1.5	−5.56	−7.34	−9.31
1.9	−5.32	−7.27	−8.71
2.7	−5.47	−G.69	−8.64
3.6	−5.47	−6.55	−8.42
4.2	−5.05	−6.47	−8.13
28	−4.89	−6.38	−7.87

**Table 4 sensors-22-01808-t004:** Comparison of existing works with the proposed antenna design.

Ref.	Bands	# of Elements	Antenna Type	Size	Bandwidth	Peak	%*η*
	(GHz)	sub-6, mm-Wave		(mm^2^)	(GHz)	Gain (dBi)	
[30]	2.45, 2.6, 5.2, 28	4, 1	Monopole, slot	1.167λ × 0.833λ	0.4, 0.5, 4	4, 11	70, 80
[31]	2.6, 2.5, 28	2, 1	monopole, slot	1.81λ × 1.81λ	0.45, 0.58, 5	4.2, 15	80, 85, 90
[32]	2.3–3.5, 5, 28, 38	1, 2	wire &; slot	1.69λ × 1.231λ	1.2, 0.75, 2, 3	2.3, 4.2, 8, 9	70–80
[33]	0.8, 2, 28	U	Monopole, Vivaldi	0.8λ × 0.4λ	0.2, 0.8, 1	4.3, 8.2	75–88
[34]	3.6, 28	1, 0	Dipole, slot	1.8λ × 0.6λ	0.75, 3	4, 8	85–90
[35]	2	1, 1	monopole	0.6λ × 0.67λ	3.2	3.2	85
Prop. work	1.5, 1.9, 2.4–2.9	2, 2	slot antenna	0.5λ × 0.35λ	0.25, 0.5	2.2–8.2	76–91

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
