# Peer review of "A Multiband Shared Aperture MIMO Antenna for Millimeter-Wave and Sub-6GHz 5G Applications"

_sensors, 2022, doi:10.3390/s22051808_

Round 1

Reviewer 1 Report

The paper is well written and seems solid in terms of its fundamental ideas. 

Nevertheless, a proofread by a native English speaker would be gainful.

The experimental results outperforms other approaches.

Please develop more future work details in the last section. You can add a future work subsection for that.

Also, you can present a concrete example to demonstrate how this system could be used in the real-world.

All in all, this study can bridge the gap between theory and practical implementation, providing us with a solution in a decent amount of time. Based on that, I recommend this paper for publication.

Reviewer 2 Report

This research article has proposed and investigated a two-element shared aperture MIMO antenna design for 5G-enabled devices and access points. It covered the sub-6GHz and millimeter-wave bands with the same radiating elements and four concentric pentagonal slots comprising the single element. The idea of the paper is interesting and it was supported with interesting results. The achievements of the paper have been experimentally validated and compared with prior arts. So, it seems the proposed work can be attractive for antenna society, however before final recommendation authors are requested to carefully revise the paper according to the provided comments to improve its quality.

1) Abstract section is short and it needs to be extended by adding more explanations on the proposed work and its achievements. For example, the design process of the proposed MIMO antenna can be discussed briefly.

2) The performance parameters of the proposed MIMO antenna such as dimensions, mutual coupling between the array elements, edge-to-edge distance between the array antennas, and radiation gain and efficiencies should be added in the abstract section.

3) At the end of the abstract section authors can mention the advantages and the practical applications of the proposed waveguide antenna array.

4) Introduction section can be improved by providing more comprehensive information, especially about the MIMO array antennas. For example, to design MIMO array antennas, the mutual coupling is a crucial issue which should be addressed by designers. There are various techniques to suppress the mutual coupling between the array antennas which can be briefly mentioned and referenced in the introduction section. Below are helpful suggestions.

“Wideband linear microstrip array antenna with high efficiency and low side lobe level” Int J RF Microw Comput Aided Eng. 2020;e22412, doi: 10.1002/mmce.22412.

"A Comprehensive Survey on "Various Decoupling Mechanisms with Focus on Metamaterial and Metasurface Principles Applicable to SAR and MIMO Antenna Systems"", IEEE Access, vol. 8, pp. 192965-193004, 2020.

"Low-Interacted Multiple Antenna Systems Based on Metasurface-Inspired Isolation Approach for MIMO Applications", Arab J Sci Eng (2021). https://doi.org/10.1007/s13369-021-05720-6.

5) Table 1 provides an interesting comparison with prior arts, however this table can be extended by adding more works such as the above mentioned suggestion. However, the terms of comparison can be extended as well by adding “the applied design method, frequency bandwidth, average mutual coupling between the array antennas, average radiation gain and efficiency, design complexity”. Also, it is better to move this table to the end of the paper before conclusion.

6) The feeding mechanism of the array antennas shown in Fig.2 should be discussed in more detail. Why are different feedings?

7) How the mutual coupling was controlled or suppressed between the array elements shown in Fig.3? Please discuss.

8) How authors have extracted the equivalent circuit model shown in Fig.9? Please discuss it in depth which can help readers to better follow the proposed work.

9) Can authors please add the 2D curves of the radiation gain and efficiency of the array antennas over the operating frequency band. 

Reviewer 3 Report

In this paper a multiband MIMO antenna for millimeter-wave and sub-6ghz 5g applications is proposed. This manuscript can be published after major revision. There are some modifications should be done.

1- The Comparison Table should be modified and others parameters like as Bandwidth, Peak gain (PG) and efficiency (%η) should be added in the comparison Table.

2- To have better comparison size of the proposed work and other works should be provided as both of the mm and λ. Because the size of circuits is depended to the wave length and orating frequency.

3- The Comparison Table should be located after result, where the results of the proposed antenna are obtained. Why reported in introduction section?

4- The Size of device is too big for 5G application and millimeter-wave can you describe this issue.

5- Provide sufficient explanations about 5G and millimeter-wave.

6- High quality figures about fabricated photo (Fig2 (a) and (b)) and fig 12 (e) should be provided.

7- Some parts of manuscript is similar to [R1] , which should be modified.

“The paper is organized as follows. The antenna design details along with antenna design procedure 147 equivalent circuit diagram and analysis are discussed in Section II. In Section III, the measured and 148 simulated scattering parameters results are presented along with the radiation characteristics of the 149 proposed antenna design. Conclusions of the proposed work are provided in Section IV. “

 8- The main contribution should be emphasized clearly?

9- Quality of all of figures should be improved

10- Peak gain (PG) and efficiency (%η) parameters for the proposed antenna should be obtained and reported.

11 – In line 407, Editing error: “$times$8 power divider” should be revised.

12- A briefly discussion about power divider design should be added, maybe below paper is helpful” Size reduction and performance improvement of a microstrip Wilkinson power divider using a hybrid design technique. Scientific Reports. 2021 Apr 8;11(1):1-5”

13- The Current density distribution diagram only reported at 1.98 GHz, 2.69 GHz, 3.6 GHz, 4.25 GHz, while in the manuscript is claimed that the proposed antenna covers frequency bands: 4-4.5 GHz, 3.1-3.8 GHz, 2.48-2.9 GHz, 1.82-2.14 GHz, 21 and 1.4-1.58 GHz, 28 GHz

[R1] Rifaqat Hussain. "Shared Aperture Slot-Based Sub-6 GHz and mm-Wave IoT Antenna for 5G Applications", IEEE Internet of Things Journal, 2021

Round 2

Reviewer 2 Report

The quality of the manuscript in its revised version has been improved by successfully addressing the reviewers' concerns. So, at this stage there are no more technical comments.  

Reviewer 3 Report

The manuscript can be accepted in present form.